# Comparison of Mechanical and End-Use Properties of Grey and Dyed Cellulose and Cellulose/Protein Woven Fabrics

**DOI:** 10.3390/ma14112860

**Published:** 2021-05-26

**Authors:** Eglė Kumpikaitė, Sandra Varnaitė-Žuravliova, Indrė Tautkutė-Stankuvienė, Ginta Laureckienė

**Affiliations:** 1Department of Production Engineering, Faculty of Mechanical Engineering and Design, Kaunas University of Technology, Studentų Str. 56, LT-51424 Kaunas, Lithuania; indre.tautkute-stankuviene@ktu.edu (I.T.-S.); ginta.laureckiene@ktu.lt (G.L.); 2Center for Physical Sciences and Technology, Department of Textiles Physical-Chemical Testing, Demokratų Str. 53, LT-48485 Kaunas, Lithuania; sandra.varnaite.zuravliova@ftmc.lt

**Keywords:** cellulose and cellulose/protein woven fabrics, finishing, washing, colour fastness, mechanical properties, end-use properties

## Abstract

The behaviour of textile products made from different fibres during finishing has been investigated by many scientists, but these investigations have usually been performed with cotton or synthetic yarns and fabrics. However, the properties of raw materials such as linen and hemp (other cellulose fibres) and linen/silk (cellulose/protein fibres) have rarely been investigated. The aim of the study was to investigate and compare the mechanical (breaking force and elongation at break) and end-use (colour fastness to artificial light, area density, and abrasion resistance) properties of cellulose and cellulose/protein woven fabrics. For all fabrics, ΔE was smaller than three, which is generally imperceptible to the human eye. Flax demonstrated the best dyeability, and hemp demonstrated the poorest dyeability, comparing all the tested fabrics. The colour properties of fabrics were greatly influenced by the washing procedure, and even different fabric components of different weaves lost their colours in different ways. Flax fibres were more crystalline than hemp, and those fibres were more amorphous, which decreased the crystallinity index of flax in flax/silk blended fabric. Unwashed flax fabric was more resistant to artificial light than flax/silk or hemp fabrics. Finishing had a great influence on the abrasion resistance of fabrics. The yarn fibre composition and the finishing process for fabrics both influenced the mechanical (breaking force and elongation at break) and end-use (area density and abrasion resistance) properties of grey and finished fabrics woven from yarns made of different fibres.

## 1. Introduction

Colour fastness to artificial light and other effects has been investigated by various scientists. The demand of numerical evaluations of textile colour appears because of the problematic establishing of shade difference between appropriate and invalid dyeing. All factors, such as the colour establishing method, the evaluation of acceptable limits, the importance of spectral harmonisation, the usage of direct and contactless colour evaluation systems, and the verification of dry, wet and finished fabrics’ colours, are very important for textile dyeing [1]. The degree of colour fastness to light can be divided into different levels, and there are many factors that can affect the colour fastness to light, such as fibre materials, dye, colour combinations, the atmospheric composition of the surroundings, sample moisture content ratio, temperature, etc. [2].

Broadbent analysed the development of reactive dyes; different principles of dyeing cotton with reactive dyes; dye reactivity, application and storage; the fastness properties of reactive dyes on cellulosic fibres; the evidence for covalent bond formation with cellulose; bifunctional reactive dyes; and reactive dyes for wool [3]. Reactive dyes are the textile dyes, which make the covalent bonds with textile material. They dye cellulose and polyamide fibres. The fabrics dyed with reactive dyes have gloss and variety of shades. They are universal in usage and have high moisture resistance [3,4].

Fergusson investigated colour fastness in relation to different finishing. Colour loss speed increased, when fabric was exposed to light wet than dry. Detergents enhanced colour loss when fabric was exposed to light. The main reasons were the oxidising bleach in the detergent and moisture. Light, temperature, moisture, alkalis, oxidising bleach, and UV radiation acted on dye desorption during washing and scouring. Pure water enlarged the colour loss, possibly because of dissolved oxygen. UV radiation and oxidising bleach changed dyes chemical structure as well as their shade, especially for dark colours (black and dark blue) [5].

Okada, Sugane, Fukuoka, and Morita established that many dyes were decoloured from the wet cotton fabric; at first, by oxidation, later, by reduction [6,7,8]. Lee et al. determined that cotton fabrics have higher colour decolouration and smoothness than hemp fabrics. The colour fastness was similar [9].

The textile dyeing process influences the crystallinity of fibres. The investigation of crystallinity of the cellulose fibres showed that higher extraction amount influences the lower crystallinity, and lower size of the crystallite can accelerate decomposition and reduces the thermal stability of cellulose fibres. Additionally, thermal decomposition of natural fibres takes place at a higher temperature, when cellulose crystallinity and crystallite size increase. The results showed that cellulose crystallite size influenced the degradation temperature. There is some possibility of obtaining information about cellulose fibre structure and properties before their usage in textile composites [10].

The organosolv pulping crystallinity degree and crystallite size were higher than soda and sulphate pulping. Not only chemical factors, but also temperature and pressure in boiling apparatus influences the crystallite cellulose structure during the boiling process [11].

The low curing temperature and low agent concentration decrease the cellulose crystallinity and crystallite size. However, crystallite size and crystallinity reduce significantly when the curing temperature is higher than 180 °C and agent concentration is higher than 6%. The reason could be the acidic erosion of crystals under high temperature. Initial low crystallinity loss influences high fabric strength loss. This can be influenced by the reason that, at first, the crystallinity decreases. The strength loss of the bonds are influenced by the reduced crystallinity between crystallites and amorphic places [12].

Crystallinity reduces during the processes of scouring and mercerization, and no significant influence appears in crystallinity in desizing or resin application processes. When the tensile strength increases, the resin application decreases [13].

The crystallinity and its effects on different kinds of wood, from which the cellulosic fibres are excluded, were also investigated in [14,15] using different methods of investigation. The colour fastness of dental cement is also a significant problem in everyday practice [16,17,18].

The breaking force and elongation at break are the main mechanical properties of textiles. Rathod and Kolhatkar [17] investigated the physical properties of fabrics from bamboo fibres and blended bamboo/cotton (50:50) yarns. These fabrics are commonly used for clothes. The results of their analysis of the properties of different fabrics showed that fabrics made from 100% bamboo displayed higher abrasion resistance than fabrics made from bamboo/cotton (50:50). In addition, the elongation at break was higher for fabrics made from 100% bamboo than for fabrics made from bamboo/cotton (50:50) [17].

Parameters of the warp and weft threads influence the fabric’s breaking force. Their properties (raw material, linear density, and structure) influence the character of the stress–strain curves, as well as the fabric breaking force in warp and weft directions [19,20]. The fabric’s breaking force and elongation at break depend on the yarns used and their breaking forces, as well as the elongation at break.

Gabrijelčič et al. [21] established that when single and plied twisted yarns are used, the fabric breaking force from plied yarns in the weft direction is higher than that from single yarns. Gunaydin et al. [22] investigated the influence of different compositions of different cotton fibre blends on the tensile properties of yarns from these blends. It was established that the composition of the cotton blend influences the properties of yarn made from these blends.

Azeem et al. [23] established that, when yarn interweaving increases, the crimp in the load bearing increases, breaking force decreases, and broad floats decrease weaving the polyester/cotton (50:50) yarn in the weft and 100% cotton in the warp. Elementary weaves were used for the fabrics. The spinning technique influences these properties significantly.

Currently, the design of high-quality textile products and investigation of their properties are highly relevant. Predictions of technological and end-use properties of woven fabrics are a very important problem in the fabric creation process. The yarn raw material, fibre fineness, yarn linear density, yarn type, yarn tensile properties and hairiness, weave, setting, etc., all influence the above-mentioned fabric properties [20,21,22,23].

Air permeability, abrasion resistance, mass loss, pilling resistance are influenced by the yarn raw material, fibre fineness, yarn linear density, yarn type, yarn tensile properties and hairiness, weave, area density, etc. [24,25,26,27,28,29].

Textile waste is used for the creation of regenerated fibres and fabrics, especially for cellulose composites. Denim fabrics are used for reinforcement material and dissolved cellulose—for the matrix phase. Tensile, impact and flexural properties of the created composites were established. The structure of manufactured textile composites was also investigated by scanning electron microscopy [30].

Thus, the behaviour of textile products from different fibres during their finishing has been investigated by many scientists, but investigations have usually been performed with cotton or synthetic yarns and fabrics. However, the properties of raw materials such as linen and hemp (other cellulose fibres) and linen/silk (cellulose/protein fibres) have rarely been investigated. The aim of this article was to investigate and compare the mechanical (breaking force, elongation at break) and end-use (colour fastness to artificial light, area density, abrasion resistance) properties of cellulose and cellulose/protein woven fabrics.

## 2. Materials and Methods

Woven fabrics made using three different raw materials—with fibres originating, to a large extent, from bast fibres—were investigated; 100% linen long fibre yarns (28 tex), as well as 100% hemp long fibre (28 tex) yarns and yarns of fibres made from 70% linen/30% silk (26 tex) blend, were used for fabrics’ warp and weft. Fabrics were woven by the joint stock company “Klasikinė tekstilė”. The weave of the fabric was combined—one-layered (dense parts, woven in plain weave) and double-layered (sparse parts, both layers woven in plain weave). The drawing-in scheme and weave cards are presented in Figure 1; Figure 2. Fabrics were woven using an Itema R9500 (Itema Group, Colzate, Italy) rapier weaving loom. The warp setting was 220 dm^−1^ and the weft setting was 200 dm^−1^. Grey and finished fabrics of all three raw materials were investigated.

Yarns were dyed in the department of yarn dyeing in the “Klasikinė tekstilė” company. Flax and hemp yarns were dyed using mass-produced yarn dyeing equipment, “KRANTZ” (Krantz GmbH, Aachen, Germany), and flax/silk yarns were dyed separately using low-capacity sample dyeing equipment, “THIES” (Thies GmbH & Co. KG, Coesfeld, Germany).

Yarns were dyed with reactive dyes at a temperature of 60 °C using sodium hydroxide (for dyeing flax/silk yarn using calcined soda). They were rinsed, neutralised (with vinegar) to reduce alkalinity, and saponified at 80 °C. They were then paraffinised (due to the better yarn properties required for the weaving process) and centrifuged. They were dried in a convectional dryer Unitech 2200 (Unitech, Montemurlo, Italy) at 80–85 °C. The dyeing process took 6–8 h, depending on colour intensity (shorter for lighter colours, longer for darker colours). Yarns were dried in the drier for 2.5 h.

Fabrics were finished. They were washed, dyed, rinsed, and softened in a Brongo 100 (Brongo srl, Florence, Italy) machine. The fabrics were washed for 10–15 min at a temperature of 65 °C. Dyeing continued for 75–120 min at a temperature of 60 °C. Warm and cold rinsing was carried out for the fabric (one rinsing took 5 min). Then, acid and softener were poured into the machine. Active dyestuffs from Everzol (Everlight Chemical, Taipei, Taiwan) were used for dyeing and softener from the Perustol CCF (Rudolph Group, Gerestried, Germany) was used for softening. It is clear that different chemical reactions take place in the finishing processes, but it is beyond the scope of this article to describe chemical processes and reactions. The aim of the investigation was to study yarn and woven fabric behaviour during the finishing process.

The surface colour properties of the tested fabrics were measured according to the EN ISO 105-J01 standard, and the colour difference was measured according to the EN ISO 105-J03 standard with a Spectraflash SF450 spectrophotometer (Datacolor, Lawrencewille, NJ, USA) coupled to a PC, in the wavelength range of 360–700 nm, geometry type d/80, with specular component included, using a 9 mm area of view, with UV energy included. Illuminant and observer conditions were described in the standard D65/100. All parameters of surface colour measurements and different colour values were measured under the same testing conditions with the same instrument. A one-layer fabric specimen with a white backing was used during the measurement procedure. In general, the colour measurement was a numerical representation of the colour of a specimen obtained by use of a colour measuring instrument; an average of four readings of a specimen were presented as a result. The principle of the method used to measure surface colour properties and colour differences of fabrics was based on reflectance methods in order to obtain a numerical representation of the colour of the specimen.

EN ISO 105-J03 standard provides a method of calculating the colour difference between two specimens of the same material, measured under the same conditions, such that the numerical value ∆Ecmc(l:c) for the total colour difference quantifies the extent to which the two specimens do not match. It permits the specification of a maximum value (tolerance) which depends only on the closeness of match required for a given end-use and not on the colour involved, nor on the nature of the colour difference. The described method enables establishing of ratio of lightness, chroma and shade difference.

An average out of 4 readings per sample was recorded as colour yield (*K*/*S*), colour difference (∆*E***_ab_*), and other colour parameters, such as *L**, indicating lightness; *a**, the red/green coordinate; and *b**, the yellow/blue coordinate. Delta values for *L** (Δ*L**), *a** (Δ*a**), and *b** (Δ*b**) may be positive (+) or negative (−):

Δ*L** (*L** sample minus *L** standard) is the difference in lightness and darkness (+, lighter; −, darker);

Δ*a** (*a** sample minus *a** standard) is the difference in red and green (+, redder; −, greener);

Δ*b** (*b** sample minus *b** standard) is the difference in yellow and blue (+, yellower; −, bluer).

The *C***h* colour space is similar to *L***a***b**, but it describes colour differently using cylindrical coordinates instead of rectangular coordinates. In this colour space, *C** represents chroma, and *h* is the hue angle. Chroma and hue are calculated based on *a** and *b** coordinates in *L***a***b**. Deltas for chroma (Δ*C**) and hue (Δ*H**) may be positive (+) or negative (−):

Δ*C** (*C** sample minus *C** standard) is the difference in chroma (+, brighter; −, duller);

Δ*H** (*H** sample minus *H** standard) is the difference in hue.

The colour difference is expressed as Δ*E***_ab_* and is calculated using the Equation (1):(1)$$∆E_{ab}^{*}=\sqrt{{\left(∆L^{*}\right)}^{2}+{\left(∆a^{*}\right)}^{2}+{\left(∆b^{*}\right)}^{2}}$$
where Δ*E***_ab_* is the CIELab colour difference between the batch and standard in commensurate units, ∆*L** denotes the difference between lightness (where *L** = 100) and darkness (where *L** = 0), ∆*a** is the difference between green (−*a**) and red (+*a**), and ∆*b** is the difference between yellow (+*b**) and blue (−*b**).

The total difference Δ*E***_ab_* is always positive.

The substrate absorption function (*K*), the scattering function of background (*S*) and the reflectance (*R*) in the visible spectrum (400–700 nm) are linked by the Kubelka–Munk equation (Equation (2)):(2)$$\frac{K}{S}=\frac{{\left(1-R\right)}^{2}}{2R}$$

The extent of apparent fixation (%) achieved was calculated using Equation (3):(3)$$F\%=100\times \frac{{\left(\frac{K}{S}\right)}_{2}}{{\left(\frac{K}{S}\right)}_{1}}$$
where (*K*/*S*)_1_ is the *K*/*S* values of dyed fabric before washing, (*K*/*S*)_2_ is the *K*/*S* values of dyed fabric after washing, and *F*% is the degree of fixation of absorbed dye.

FTIR spectra were obtained using a Perkin-Elmer PE Spectrum-BX FT-IR Infrared spectrometer (PerkinElmer Inc., Cleveland, OH, USA). The spectra were recorded in a 400–4000 cm^−1^ spectral range with a 4 cm^−1^ resolution. The number of scans applied was 32. The specimens were prepared under similar conditions. The specimen was ground in an agate mortar and mixed with both exclusively pure and purified-for-analysis KBr powders, then placed in a mould and compressed under high pressure into a transparent thin disk. The weight of the specimen was approximately 0.202 mg.

The crystallinity index of the bast fabrics was calculated according to Equation (4):(4)$$k_{i}=\frac{a}{b}$$
where *a*/*b* is the ration peak intensity, ranging from 1163 cm^−1^ to a peak of 1159 cm^−1^, which was determined using the baseline method (see Figure 3).

The conditioning of samples before testing was carried out for 24 h in a standard atmosphere: temperature 20 °C ± 2 °C, relative humidity 65% ± 4%, according to the EN ISO 139 standard.

The colour fastness to washing tests were carried out according to the EN ISO 105-C06 standard, test methods A1S and A1M, at 40 °C, using 10 metal balls. The apparatus used was the SCOUROTESTER FE–09/A (Komputekst, Szekesfehervar, Hungary); the detergent used was ECE Reference Detergent with Phosphates without an optical brightener; and the adjacent fabric used was Multifibre, type DW. Scouring finishing in acetic acid reagent was not conducted after the washing procedure. A visual assessment was carried out according to EN 20105-A02 and EN 20105-A03. According to the standard EN ISO 105-C06, the colour loss and staining appearing due to desorption and (or) abrasion during one (A1S) test matches one industrial or domestic washing cycle. The results of one multiple test (A1M) correspond to five domestic or industrial washing results. M tests are more compound than S tests because of higher mechanical action. In other words, the times of washing procedures of these two methods were different: when the test was carried out according to the A1S method, specimens were washed for 30 min, and for the A1M method, the washing took 45 min.

Samples without pre-finishing and after colour fastness to washing tests were subjected to xenon testing in the Megasol V2.01 apparatus (James H. Heal & Co, Halifax, UK)—the colour fastness to artificial light was measured according to the EN ISO 105-B02 standard, method 1. Exposure conditions: 300–400 nm intensity of irradiation, 92 W/m^2^, 29 °C temperature, and 32% humidity.

A visual assessment was carried out according to EN 20105-A02 and EN 20105-A03 standards and an instrumental assessment according to EN ISO 105-A04 and EN ISO 105-A05 standards for the change in colour and staining of adjacent fabrics, respectively.

An abrasion resistance test was performed according to the LST EN ISO 12945-2 standard using a MESDAN-LAB, Code 2561E (SDL Atlas, Rock Hill, UK) Martindale abrasion and pilling tester. The tester was stopped after every 1000 abrasion cycles until two threads broke on the fabric surface. The mass of the fabric samples was fixed after each abrasion period.

The area density of cut samples was established in accordance with the ISO 12,127 international standard. Fabric samples of 100 cm^2^ were weighed using an EW 150-3M electronic balance (Kern and Sohn GmbH, Balingen, Germany).

The main mechanical properties of yarns were established using a universal stretching machine, “Zwick/Roell” (ZwickRoell GmbH & Co.KG, Ulm, Germany). The stretching machine was connected to a computer, on which the special software testXpert^®^ (ZwickRoell GmbH & Co.KG, Ulm, Germany) was installed. The tests were performed according to the standard LST EN ISO 2062:2010. The active length of the yarns was 500 mm, and the stretching speed was 500 mm/min. The device used a 50 N force measuring sensor. The accuracy of tensile machine measurements was 0.01 cN for force and 0.01% for elongation.

The breaking force and elongation at break for grey and finished fabrics were established using a Zwick/Z005 (ZwickRoell GmbH & Co.KG, Ulm, Germany) universal tensile machine, according to standard ISO 13934-1:1999.

Five strips from each investigated fabric were cut. The width of the strips was 60 mm; 5 mm fringes were left in each selvedge. The sample was fixed over its whole width into the clamps of the tensile machine, where one clamp was stationary, and the other clamp moved at a constant speed until the sample broke during the test. The work length of the samples was 200 mm ± 1 mm. Breaking force and elongation at break were established during the test. The arithmetic average, standard deviation, and coefficient of variation were calculated. Tensile tests were performed only in the warp direction.

## 3. Results

Two kinds of cellulose and one kind of cellulose/protein raw woven fabric of a plain weave, with dense and spare parts, were chosen for investigations. Both parts of the plain weave had different settings; therefore, the colour differences between those parts in the fabric was measured, and are presented in Table 1. The data showed that colour differences in the fabrics were very small, i.e., less than two, and could not be detected by the human eye. However, human eyes do not reflect shade, chroma and lightness equally, i.e., the acceptance limits of different shades differ [1]. When ∆*E* is equal to 1.0, the colour difference is the smallest the human eye can see. If ∆*E* is lower than 1.0, the shade is not noticeable. However, sometimes when ∆*E* is lower than 1.0 it can be unnoticeable; for example, two yellows of the same hue, whose ∆*E* is lower than 1.0, cannot be observed. This appears due to the saturation of yellow colour [5]. Usually, according to ISO standard, when 1 < Δ*E* < 2, there are small changes, which can be perceived only by an experienced observer. In many investigations, ∆*E* < 3 is acceptable by every human eye [16,17,18]. Colours that fade off-tone are more easily distinguished that those that fade in-tone. When the colour changes because of fading, it is significantly noticeable. If the fade is in the same shade, but different depth, it can be noticed less easily [5].

The grey and dyed fabrics are shown in Figure 4.

Reactive dyes include chromophore and reactive groups. Therefore, they have good moisture resistance due to covalent bonds with cellulose fibre [4]. The colour can leach from the fabric to water during domestic washing. The fabric colour can change due to colour loss or colour change can be tonal (i.e., on-tone), or the colour can change in shade (i.e., off-tone) [5]. The colours of all tested fabrics did not change significantly; for example, the evaluation of grey scale is 4–5, and the colours of neighbouring fabrics did not change. These results prove that only minimally hydrolysed dyes were on the fabric surface [5]. Mechanical influence during washing facilitated a removal of hydrolysed dyes, which separated because hot water destabilised the relationship between dyes and fibres. However, the impact of water on colour loss was low. Usage of detergent during the washing enhanced colour loss and shade differences [5].

The measured values of CIE *L***a***b** coordinates and *K*/*S* at a maximum wavelength of 480 nm are presented in Table 2. The results show that the *K*/*S* values of pure flax fabric were the highest and the *K*/*S* values of pure hemp fabric were the smallest. This suggests that flax has the best dyeability and the hemp has the poorest dyeability, compared to all the tested fabrics. The reason for these results may be due to the different amounts of cellulose and lignin in flax and hemp fibres. Hemp fibre contains 5.68–7.96% lignin and 64.7–75.38% cellulose; flax fibre has 75.38–83.31% cellulose and 2.21–4.78% lignin [27]. Hemp yarns are rougher and have more remnants of lignin than flax yarns [31]. Thus, these reasons may influence fabric dyeability, although chemical tests were not performed during this investigation. This is a problem of chemical technology of fibrous materials, and we have analysed the end-use properties of textile fabrics from these fibres. However, the flax/silk fabric had the medium value, because silk is a fibre of a different nature in the fabric and its dyeability with reactive dyes is worse than the dyeability of cellulose fabrics. Salt, which is used in the dyeing process of fabrics, influences *K*/*S* values as well [9]. It has been shown in the literature that the diffusion coefficient of dye is a function of both the dye and electrolyte (salt) concentrations [3]. The dye exhaustion of reactive dye is a linear function of the salt concentration within the cellulose in the presence of a fixed amount of added dye, although the slope decreases with an increase in salt concentration [9].

It can clearly be seen in Table 3 that the hues of all the tested fabrics became different after washing. It can be stated that the colour properties of fabrics are greatly influenced by the washing procedure, and even the fabric parts of different weaves lost their colour in different ways, e.g., flax fabric became lighter after washing, but the dense parts of the fabric were distinguished by a brighter colour with redder and yellower tones, whereas the sparse parts of the fabric had greener and bluer tones and the colour became much duller after washing. The dense parts of flax/silk blended woven fabric became lighter with brighter redder tones, whereas the sparse parts of this fabric became darker after washing. The colour of the pure hemp sample turned out to be lighter with duller blue tones. Such contrasts in the colour of bast-woven fabrics after washing may be explained not only by the dyeability of the fibres, but by their different supermolecular structures and crystallinity [9]. The results pertaining to colour differences, presented in Table 3, indicate that fabrics change their colour after washing. Regarding the literature analysis, the ∆*E* was approximately equal to one, which suggests that the change in colour may be seen by the human eye. The biggest change in colour was for hemp fabric after washing using procedure A1M (which represents five domestic washing cycles at 40 °C), because the ∆*E* in this case was greater than two. Nevertheless, it still can be stated as an acceptable result, depending on the application of hemp fabric.

The values of F% (shown in Table 3) show the fixation of absorbed dyes in the fabric. These values are influenced by the concentration of added alkali during the dyeing process [9]. It can clearly be seen that the best fixation of dyes was for flax/silk-blended woven fabrics (reaching 100%), whereas the F% of hemp and flax was around 90%.

It can be seen in Table 4 that the crystallinity indices of all the tested fabrics vary from 1.601 for hemp to 1.831 for flax. The values of F% correspond to these results. The degree of crystallinity, the apparent lateral crystallite size, part of internal crystallite bond and cellulose fraction, increases when the wood density rises [14]. The results mismatch with the results of current investigation, because hemp fibre is the roughest, but its crystallinity index is the lowest. Flax/silk fibre is the most flexible from these three fibres, but its crystallinity index is the medium value. However, we can still state that flax fibres are more crystalline than hemp and that silk fibres are more amorphous and decrease the crystallinity index of flax in flax/silk blended fabric.

The FTIR spectra for the tested dyed fabrics are shown in Figure 5. The strong broad band at approximately 3500 cm^−1^ represents different stretching modes of O-H, and another, at approximately the 2900 cm^−1^ band, represents asymmetric and symmetric methyl stretching groups in the spectra of all cellulose fibres. The bands in 1640, 1430 and 1270 cm^−1^ represent different lignin groups, C=O, and C-O stretching and bending vibrations, respectively. The bands in 1400, 1163, 1059 and 1030 cm^−1^ correspond to C=O, C-H, C-O-C and C-O deformations or stretching vibrations of different groups in carbohydrates, respectively [10]. Characteristic bands corresponding to allomorphs of crystal cellulose are also in the region of 3350–3293 cm^−1^ [14]. The relative values of lignin/carbohydrate IR bands for wood of different kinds decrease, when medium wood density increases and lignin amount decreases [14]. These results prove our results, because a blend of flax/silk fibres is the rarest and the lignin amount is the lowest. A band of approximately 1430 cm^−1^ is related to the amount of crystal structure of cellulose fibres, and a band of 897 cm^−1^ represents the amorphic cellulose area [15].

The colour fastness to artificial light of the dense and sparse parts of flax woven fabric was equal to grey scale grade four, whereas for flax fabric, after washing with A1S and A1M, the colour fastness of dense and spare parts was equal to grade 3–4. The colour fastness of the dense parts of unwashed and A1S-washed flax/silk woven fabric was equal to grade 3–4, whereas that of the sparse parts was equal to grade four, and the colour fastness of both parts of flax/silk A1M-washed fabric was equal to grade 3–4. The colour fastness to artificial light of hemp unwashed and washed woven fabric in the dense and spare parts was the same: equal to grade 3–4. Therefore, it can be concluded that unwashed flax fabric is more resistant to artificial light than flax/silk or hemp fabrics. The resistance to artificial light by flax/silk fabrics is decreased due to the presence of silk. It can also be seen in the sample that silk fibres were not dyed properly, which resulted in some white fibres being stuck in the fabric. We learned that the colour fastness to light of natural bast fabrics is flax > flax and silk blending > hemp. Ha and colleagues [2] established that colour fastness depends on the fibre material, dye, colour, moisture, temperature, etc. Ha’s research confirms possible reasons for our result, which can be achieved because of the different morphology and structure of the fibres. Ingamells [19] stated that fastness to light is a function of colour depth. ∆*E* values show the total colour difference and colour differences between fabrics, but not the strength loss. The UV radiation of light source, heat, moisture, alkalis and oxidising bleach provokes hydrolysis of dye and fibre bond, causing the desorption of dye during washing and scouring [5]. *K*/*S* values presented in Table 4 shows the general colour loss, including demands due to dye desorption and light fading.

It can be seen in Figure 6 that all tested fabrics displayed a two-step fading behaviour: an initial fading after 2 h of exposure to light, and a smaller subsequent one. The same tendency was noticed by Okada et al. [6,7,8] when investigating the fastness to light of some reactive dyes. The large area density influences the initial cotton fading [6]. The possible reason of such behaviour could also be that light attacks more easily when large glue aggregates are present on the fabric surface. This influences the additional fading in the beginning of the exposition and tonal changes can be influenced by the degree of aggregation of the dye molecules in the structure of microfibre [5]. It can also be clearly seen in Figure 3 that the speed of colour fading in hemp fabrics, especially in sparse parts, after approximately 26 h of exposure to light slowly decreases, whereas the fading in other fabrics is more or less exponential.

The breaking force and elongation at break of the yarns from which the woven fabrics were made were investigated. A diagram of the breaking force of dyed and grey yarns is shown in Figure 7. It can be seen in the diagram that hemp yarns were the strongest—the breaking force of grey yarns was 7.435 N, and that of dyed yarns was 6.931 N. Linen/silk yarns were the weakest—the breaking force of grey yarns was 2.021 N, and that of dyed yarns was 3.032 N. Linen and hemp yarns were weakened by 9% and 7% after dyeing, respectively, whereas linen/silk yarns became 50% stronger. This may have been influenced by the fact that yarns with elastic fibre stretched more after dyeing, its twist increased, and the yarn thus became stronger. The tensile strength and elongation at break of silk dyed yarn are higher than that of grey yarn. On the other hand, the less elastic yarns were affected more by mechanical and chemical effects, and this weakened the yarns. As mentioned by different authors, yarns’ structure [2] and fibre composition [9], and the spinning technology used [20] have an influence on the mechanical properties of fabric made from these yarns. Previous authors [9] confirmed that the cotton/hemp blend exhibited higher elongation values and better build-up property than cotton due to the lower crystallinity. The results [9] confirm our research, because the hemp yarns had the lowest crystallinity and the highest breaking force. Thus, it is very important to investigate the mechanical properties of yarns before analysing the mechanical properties of fabrics composed of these yarns.

A diagram of the yarns’ elongation at break is shown in Figure 8. As can be expected, the most elastic yarns were yarns with elastic fibre silk (the elongation at break of grey yarns was 1.69%, and that of dyed yarns was 3.2%). The lowest elongation at break (for grey yarns—1.69%, for dyed yarns—1.79%) was recorded for linen yarns. The elongation at break of all yarns after dyeing increased from 5.5% for linen yarns to 39% for linen/silk yarns. Breaking force and elongation at break depend more on yarn structure than on air pressure in the texturing jet in textured polypropylene yarns [20]. This confirms that yarn structure influences tensile properties of the yarns. Fibre length, fineness, strength and maturity affect yarn tensility, evenness, imperfections and hairiness, which influences the yarn properties [22].

The breaking force and elongation at break of fabrics was established during our investigation. It can be seen in Figure 9 that the hemp yarn was the strongest (974.21 N for grey yarn and 470.03 N for dyed yarn). In the case of grey fabrics, linen/silk fabric was the weakest (422.93 N), but this subsequently changed by only 5%. By contrast, the breaking force changed by 58% after finishing and, in the case of finished fabrics, the linen fabric was the weakest. In the case of all fabrics, the breaking force became weaker. The breaking force of all raw materials was higher than that recorded after finishing, i.e., fabric weakened after finishing. The reason for this is that fabric is affected mechanically and chemically during the finishing process. The maximum effect of finishing in terms of breaking force was seen in linen (58%) and hemp (52%) fabric. However, fabric from linen/silk blended yarn in terms of breaking force remained almost unchanged, i.e., it was weakened by only 5%. The breaking force and elongation at break are influenced by raw material, linear density, yarn structure, yarn tensile properties and hairiness, weave, setting, etc. [20,21,22,23]. However, the main investigated parameter was the raw material of fabrics in this article. When yarns of different structures manufactured with different yarn texture parameters were used for the weft, breaking force and elongation at break differed by up to 25% [20]. The influence of yarn structure was also investigated during the investigation. The results showed that not only raw material, but mechanical properties of the weft also influenced tensile properties of the fabrics [21].

A diagram of the fabrics’ elongation at break is shown in Figure 10. It can be seen that the tendencies were different from those of the breaking force. The lowest elongation at break (9.17% for grey fabrics and 16.85% for finished fabrics) was observed for linen fabrics. The largest elongation at break (11.89% for grey fabrics and 29.76% for finished fabrics) was observed for linen/silk fabrics, because silk adds elasticity to yarn and fabric. The elongation at break after finishing of all investigated fabrics was higher in comparison to grey fabrics. This can be explained by the fact that the fabric’s threads relax and crimp more after finishing. Due to this stretching, the threads of the fabric straighten out at first, and only then does the fabric itself begin to stretch. The fabrics’ elasticity after finishing increased for this reason. The largest change in the elongation at break (60%) between grey and finished fabric was observed for linen/silk fabric. Linen and hemp fabrics displayed a similar result in terms of the change (46% and 31%, respectively), and this was very different from linen/silk fabric. It can be noted that the use of 30% of a different type of fibre (silk) can change the behaviour of fabrics, in terms of the bast fibres. The mechanical and surface properties of woven fabrics from polyester/cotton (50:50) weft and 100% cotton warp were investigated. The statistical analysis presented the significant influence of spinning technique and design to these properties [23,24]. When the ratio of crystalline and amorphic region rises, the rigidity of cellulose fibres also increases, and the flexibility of bast fibres decreases [11]. Table 4 presents crystallinity index values of the dyed fabrics. The crystallinity index could be calculated for every fabric. It is clearly seen that crystallinity index is coherent to the elongation at break of yarns or fabrics (see Figure 8 and Figure 10). Higher crystallinity index corresponds to lower elongation at break values. Higher crystallinity index shows that there are fewer amorphous areas in the fibres.

The fabric’s area density is an important factor for all double-layer fabrics. Lighter fabrics are preferred. The results of the area density of grey and finished fabrics are presented in Figure 11.

It can be seen in the diagram that the area density of all the fabrics increased after finishing—because fabrics relax, their threads converge and become denser, and the area density increases as well. The area density of linen/silk fabrics changed most significantly (36%) after finishing. The silk fibre in this fabric’s composition gives the fabric extra elasticity, which leads to greater fabric crimping and to a more significant increase in area density.

Abrasion resistance is one of the most important end-use properties for fabrics. Fabrics’ appearance changes due to abrasion. The fabric surface was plane, without any protruding or broken fibres, pills, or holes before abrasion. The fabric surface became covered by fluff, from which pills covered a larger and larger part of the fabric surface during the abrasion process. The fabrics’ yarn fibres started to break, and holes appeared later in the abrasion process. Fabric breaks when two holes form in the specimen. The results relating to abrasion resistance are shown in Figure 12. Abrasion resistance, mass loss, and pilling resistance are influenced by the yarn’s raw material, the yarn’s linear density, the weave of the fabric, the area density, etc. [24,25,26,27,28,29]. Abrasion and pilling resistance depend on yarn structure and the spinning method of cotton yarn [25]. The structural differences of cotton ring and compact spun yarns had a high influence on fabric properties. The fabrics woven from compact yarns had higher tensile strength and abrasion and pilling resistance than the fabrics from ring-spun yarns [27]. The raw material and yarn structure on fabric abrasion and pilling resistance described in this article showed similar tendencies.

Grey fabrics of all three kinds broke up at 8000 abrasion cycles. Finished fabrics were more resistant to abrasion. Finished fabrics made of hemp fibre broke up at an average of 9000 abrasion cycles, linen finished fabrics broke up at the limit of 12,000 abrasion cycles, and fabric made of blended linen/silk yarns broke up at 16,000 abrasion cycles.

The specimen mass and its changes during abrasion were analysed as well. The results of this area of study are presented in Figure 13.

It can be seen that the mass of all specimens decreased during abrasion, because the wear separated the fluff, the pills, and other parts of the fabric, which lightened the fabric. Linear equations described the change in mass sufficiently well (the determination coefficients changed from 0.951 to 0.987). The tendencies of relative mass loss and abrasion cycles of coated fabrics were overviewed. The results correspond to the nonlinear function approach [29].

Of course, the dyeing process with reactive dyes is based on chemical reactions between dyestuff and fibre. Thus, it can have some influence on the mechanical properties of woven fabric, although these changes are an object of investigation fibrous material chemical technology. We just analysed the mechanical behaviour of textiles.

## 4. Conclusions

For all fabrics, ∆*E* was smaller than three, which is generally imperceptible to the human eye, and there was no change in the staining colours of adjacent fabrics.

The *K*/*S* value of pure flax fabric was the highest, and the *K*/*S* value of pure hemp fabric was the lowest. This suggests that flax had the best dyeability and hemp had the poorest dyeability, from comparing all the tested fabrics.

The colour properties of fabrics are greatly influenced by the washing procedure. Fabrics change their colour after washing.

Flax fibres were more crystalline than hemp and silk fibres were more amorphous, which decreased the crystallinity index of flax in flax/silk blended fabric.

Unwashed flax fabric was more resistant to artificial light than flax/silk or hemp fabrics. The resistance to artificial light of flax/silk fabrics decreased due to the presence of silk. All the tested fabrics displayed a two-step fading behaviour: an initial fading after 2 h of exposure to light, and a smaller subsequent fading.

Fabric area density increased from 18% for flax fabric to 36% for flax/silk fabric after finishing. Fabrics become heavier, but softer—their hand became better.

It was established that finishing had a great influence on the fabrics’ abrasion resistance. Grey fabrics collapsed after 8000 abrasion cycles, whereas finished fabrics collapsed between 9000 and 16,000 abrasion cycles. Flax/silk fabrics were the most resistant to abrasion, and hemp fabrics collapsed first. Linear equations described dependencies exactly, i.e., when the number of abrasion cycles increases, the mass loss also increases.

The yarns’ fibre composition and the finishing process influenced the mechanical (breaking force and elongation at break) and end-use (area density, abrasion resistance) properties of grey and finished fabrics woven from yarns of different fibres.

It was established that fabrics from bast fibre yarns, i.e., 100% linen and hemp yarns, were similar in their behaviour and investigated results, although blends of cellulose fibre with protein fibre (silk) of other nature changed the fabric properties significantly.

## Figures and Tables

**Figure 1 materials-14-02860-f001:**
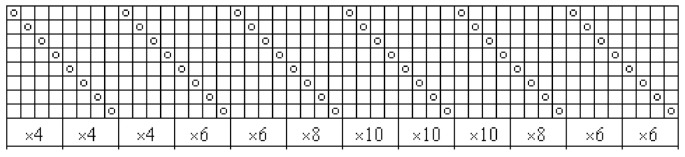
Drawing-in scheme of woven fabrics.

**Figure 2 materials-14-02860-f002:**
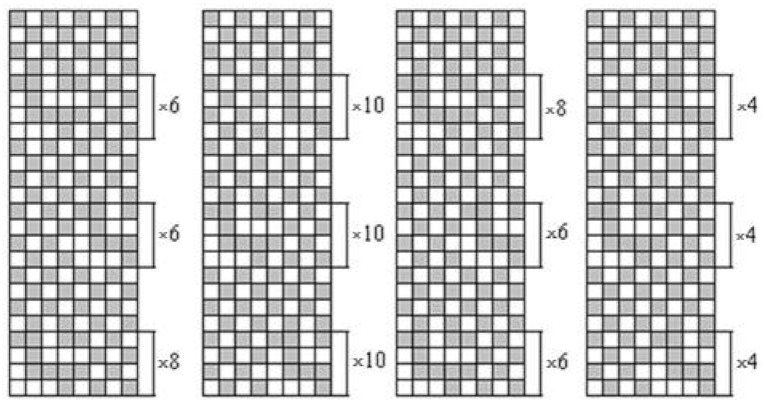
Cards of woven fabrics.

**Figure 3 materials-14-02860-f003:**
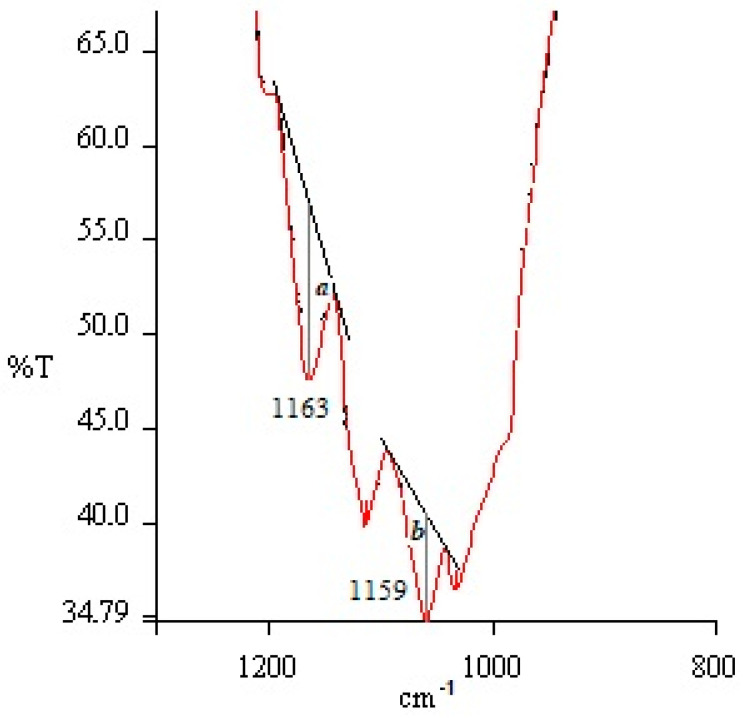
Calculation method for crystallinity index in relation to changes in absorption peaks 1163/1159 cm^−1^ in the infrared absorption spectrum.

**Figure 4 materials-14-02860-f004:**
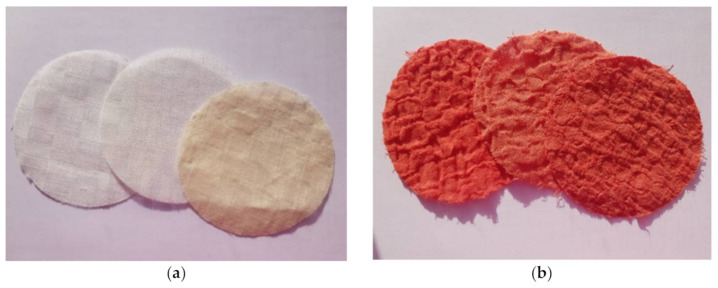
(**a**) Grey fabrics (flax, hemp, flax/silk), and (**b**) finished fabrics (flax, hemp, flax/silk).

**Figure 5 materials-14-02860-f005:**
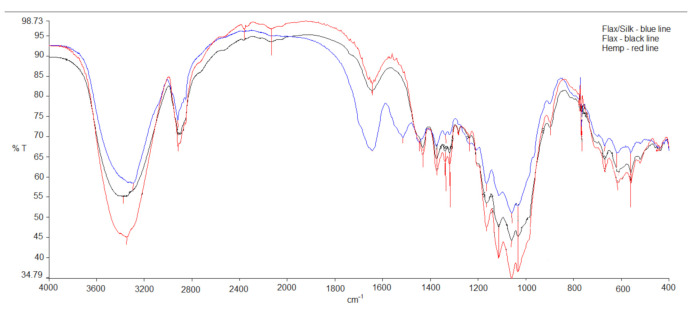
FTIR spectra of dyed bast fabrics.

**Figure 6 materials-14-02860-f006:**
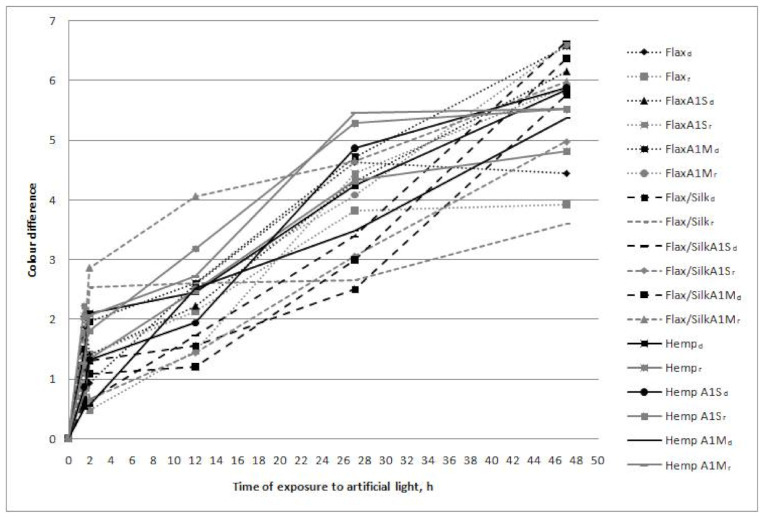
Relationship between ∆*E*_ab_* and the time of exposure to artificial light for dyed unwashed and washed woven fabrics.

**Figure 7 materials-14-02860-f007:**
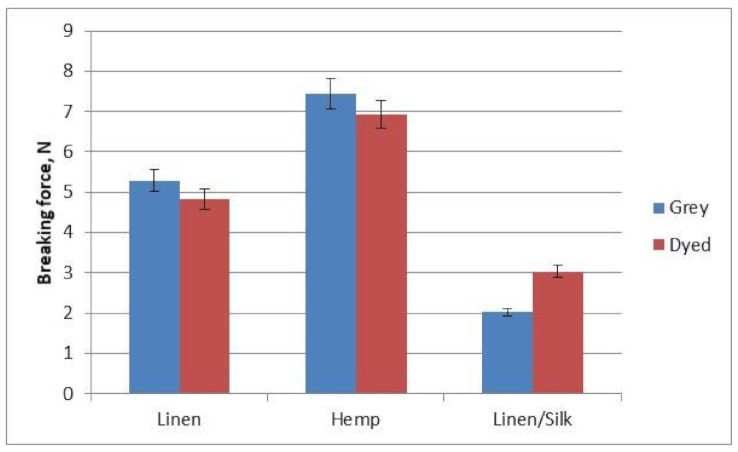
Diagram of the yarns’ breaking force.

**Figure 8 materials-14-02860-f008:**
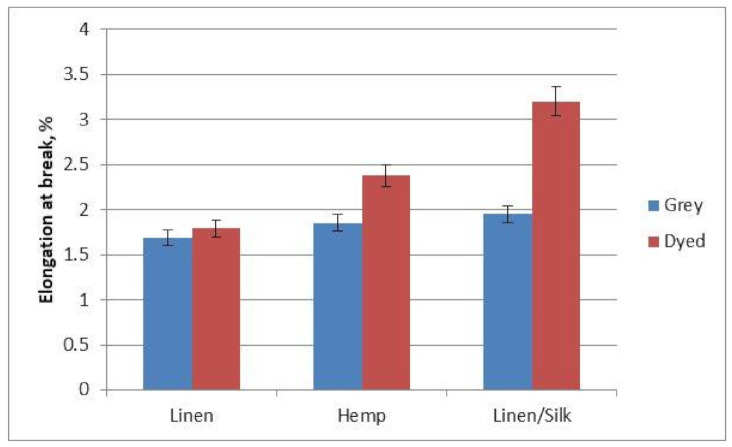
Diagram of the yarns’ elongation at break.

**Figure 9 materials-14-02860-f009:**
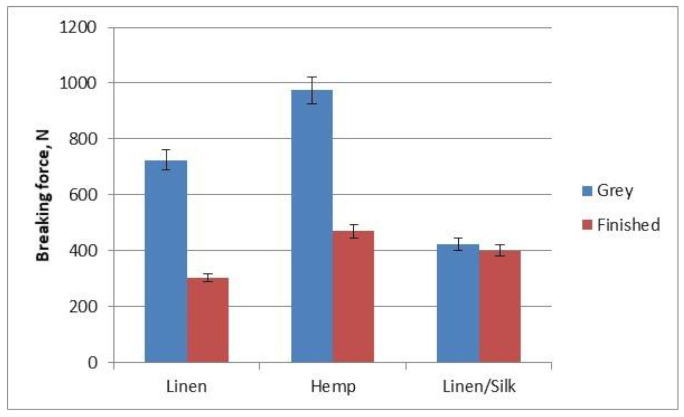
Diagram of the fabrics’ breaking force.

**Figure 10 materials-14-02860-f010:**
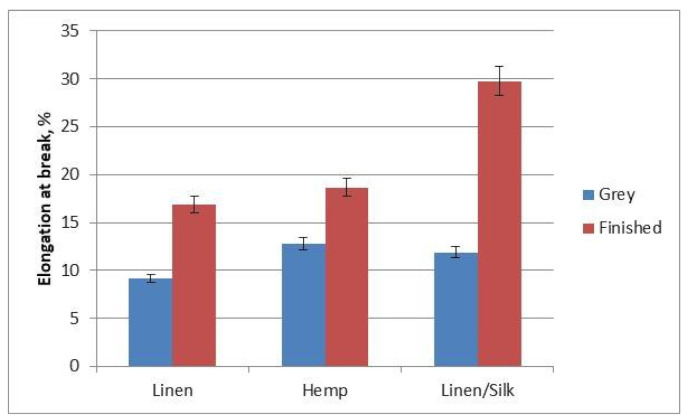
Diagram of the fabrics’ elongation at break.

**Figure 11 materials-14-02860-f011:**
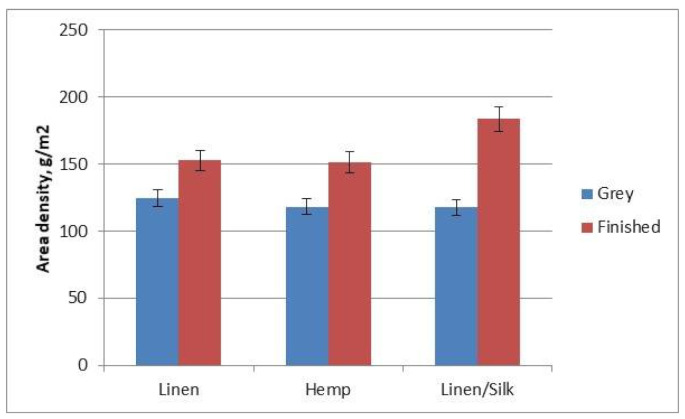
Diagram of area density.

**Figure 12 materials-14-02860-f012:**
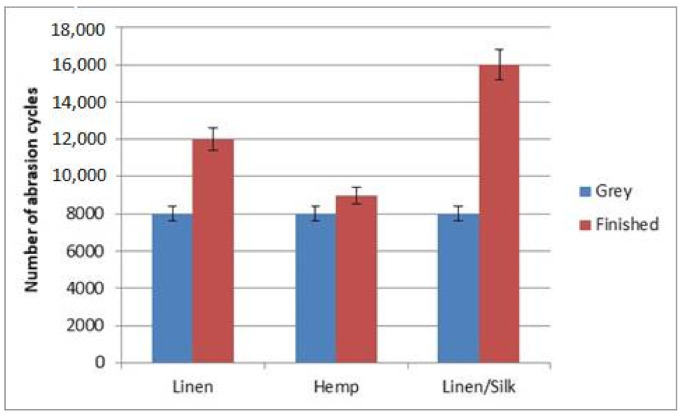
Diagram of abrasion resistance.

**Figure 13 materials-14-02860-f013:**
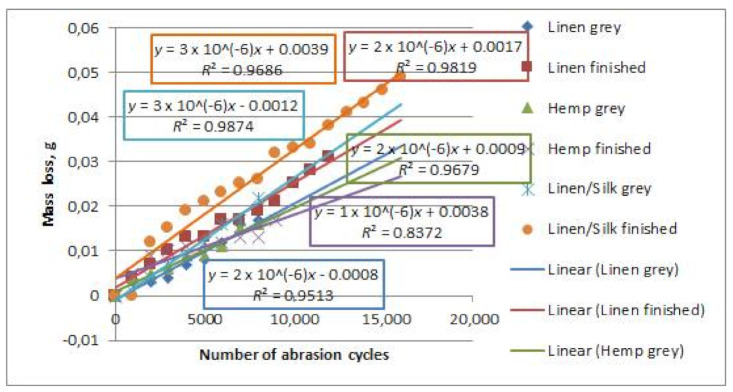
Dependencies of mass loss on the number of abrasion cycles of flax, hemp and flax/silk woven fabrics.

**Table 1 materials-14-02860-t001:** Colour difference between dense and sparse parts of dyed woven fabrics.

Code of Sample	∆*E***_ab_*, 95% Confidence Interval
Flax_d_	1.772 ± 0.009
Flax_r_	
Flax/Silk_d_	1.191 ± 0.002
Flax/Silk_r_	
Hemp_d_	1.074 ± 0.005
Hemp_r_	

**Table 2 materials-14-02860-t002:** The CIE *L** *a** *b** colour coordinates and maximum *K*/*S* values of dyed unwashed and washed fabrics.

Code of Sample	L*	a*	b*	C*	h	K/S (λmax = 480 nm)
Flax_d_	49.65 ± 0.07	43.22 ± 0.07	26.30 ± 0.07	50.59 ± 0.07	31.33 ± 0.07	6.18 ± 0.07
Flax_r_	48.30 ± 0.07	44.20 ± 0.07	26.90 ± 0.07	54.74 ± 0.07	31.32 ± 0.07	6.88 ± 0.07
FlaxA1S_d_	49.99 ± 0.07	43.41 ± 0.07	26.88 ± 0.07	51.06 ± 0.07	31.76 ± 0.07	5.98 ± 0.07
FlaxA1S_r_	48.84 ± 0.07	43.12 ± 0.07	26.44 ± 0.07	50.58 ± 0.07	31.51 ± 0.07	6.51 ± 0.07
FlaxA1M_d_	50.10 ± 0.07	43.23 ± 0.07	26.63 ± 0.07	50.78 ± 0.07	31.63 ± 0.07	5.44 ± 0.07
FlaxA1M_r_	48.43 ± 0.07	43.74 ± 0.07	25.96 ± 0.07	50.00 ± 0.07	31.28 ± 0.07	6.01 ± 0.07
Flax/Silk_d_	49.37 ± 0.07	36.95 ± 0.07	22.17 ± 0.07	43.09 ± 0.07	30.97 ± 0.07	5.00 ± 0.07
Flax/Silk_r_	48.60 ± 0.07	37.74 ± 0.07	22.62 ± 0.07	44.00 ± 0.07	30.94 ± 0.07	4.99 ± 0.07
Flax/SilkA1S_d_	49.56 ± 0.07	37.50 ± 0.07	22.25 ± 0.07	43.60 ± 0.07	30.68 ± 0.07	5.00 ± 0.07
Flax/SilkA1S_r_	47.92 ± 0.07	36.60 ± 0.07	21.55 ± 0.07	42.47 ± 0.07	30.48 ± 0.07	5.12 ± 0.07
Flax/SilkA1M_d_	50.40 ± 0.07	37.13 ± 0.07	21.99 ± 0.07	43.15 ± 0.07	30.64 ± 0.07	4.69 ± 0.07
Flax/SilkA1M_r_	47.60 ± 0.07	38.61 ± 0.07	23.11 ± 0.07	45.00 ± 0.07	30.90 ± 0.07	5.76 ± 0.07
Hemp_d_	55.37 ± 0.07	38.36 ± 0.07	24.30 ± 0.07	45.51 ± 0.07	32.35 ± 0.07	3.56 ± 0.07
Hemp_r_	54.76 ± 0.07	39.17 ± 0.07	24.64 ± 0.07	46.28 ± 0.07	32.17 ± 0.07	3.51 ± 0.07
Hemp A1S_d_	55.34 ± 0.07	39.28 ± 0.07	24.29 ± 0.07	45.34 ± 0.07	32.39 ± 0.07	3.28 ± 0.07
Hemp A1S_r_	54.85 ± 0.07	37.77 ± 0.07	24.08 ± 0.07	44.79 ± 0.07	32.52 ± 0.07	3.37 ± 0.07
Hemp A1M_d_	56.12 ± 0.07	37.64 ± 0.07	23.75 ± 0.07	44.51 ± 0.07	32.25 ± 0.07	3.24 ± 0.07
Hemp A1M_r_	56.12 ± 0.07	37.31 ± 0.07	23.68 ± 0.07	44.19 ± 0.07	32.40 ± 0.07	3.15 ± 0.07

Note: all values are presented with 95% confidence intervals.

**Table 3 materials-14-02860-t003:** The differences of CIE *L** *a** *b** colour coordinates and a degree of fixation of the absorbed dye of unwashed and washed fabrics.

Code of Sample Pair	∆L*	∆a*	∆b*	∆C*	∆h	∆E*_ab_	F%
Flax_d_	0.34 ± 0.02	0.19 ± 0.02	0.58 ± 0.02	0.47 ± 0.02	0.43 ± 0.02	0.70 ± 0.02	96.76 ± 0.92
FlaxA1S_d_							
Flax_d_	0.45 ± 0.02	0.01 ± 0.02	0.33 ± 0.02	0.19 ± 0.02	0.30 ± 0.02	0.56 ± 0.02	88.03 ± 0.88
FlaxA1M_d_							
Flax_r_	0.54 ± 0.02	−1.08 ± 0.02	−0.46 ± 0.02	−4.16 ± 0.02	0.19 ± 0.02	1.29 ± 0.02	94.62 ± 0.90
FlaxA1S_r_							
Flax_r_	0.13 ± 0.02	−0.46 ± 0.02	−0.94 ± 0.02	−4.74 ± 0.02	−0.04 ± 0.02	1.05 ± 0.02	87.35 ± 0.90
FlaxA1M_r_							
Flax/Silk_d_	0.19 ± 0.02	0.55 ± 0.02	0.08 ± 0.02	0.51 ± 0.02	−0.29 ± 0.02	0.59 ± 0.02	100.00 ± 0.91
Flax/SilkA1S_d_							
Flax/Silk_d_	1.03 ± 0.02	0.18 ± 0.02	−0.18 ± 0.02	0.06 ± 0.02	−0.33 ± 0.02	1.06 ± 0.02	93.80 ± 0.92
Flax/SilkA1M_d_							
Flax/Silk_r_	−0.68 ± 0.02	−1.14 ± 0.02	−1.07 ± 0.02	−1.53 ± 0.02	−0.46 ± 0.02	1.70 ± 0.02	102.60 ± 0.92
Flax/SilkA1S_r_							
Flax/Silk_r_	−1.00 ± 0.02	0.87 ± 0.02	0.49 ± 0.02	1.00 ± 0.02	−0.04 ± 0.02	1.41 ± 0.02	115.43 ± 0.93
Flax/SilkA1M_r_							
Hemp_d_	−0.03 ± 0.02	0.92 ± 0.02	−0.01 ± 0.02	−0.17 ± 0.02	0.04 ± 0.02	0.92 ± 0.02	92.13 ± 0.92
Hemp A1S_d_							
Hemp_d_	0.75 ± 0.02	−0.72 ± 0.02	−0.55 ± 0.02	−1.00 ± 0.02	−0.10 ± 0.02	1.38 ± 0.02	91.01 ± 0.92
Hemp A1M_d_							
Hemp_r_	0.09 ± 0.02	−1.40 ± 0.02	−0.56 ± 0.02	−1.49 ± 0.02	0.35 ± 0.02	1.51 ± 0.02	96.01 ± 0.92
Hemp A1S_r_							
Hemp_r_	1.36 ± 0.02	−1.86 ± 0.02	−0.96 ± 0.02	−2.09 ± 0.02	0.23 ± 0.02	2.49 ± 0.02	89.74 ± 0.89
Hemp A1M_r_							

Note: all values are presented with 95% confidence intervals.

**Table 4 materials-14-02860-t004:** Crystallinity index values of bast fabrics.

Sample	Crystallinity Index ki, 95% Confidence Interval
Flax	1.831 ± 0.009
Flax/Silk	1.667 ± 0.006
Hemp	1.601 ± 0.004

## Data Availability

Data is contained within this article.

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
