# Peer review of "Comparison of Mechanical and End-Use Properties of Grey and Dyed Cellulose and Cellulose/Protein Woven Fabrics"

_materials, 2021, doi:10.3390/ma14112860_

Round 1

Reviewer 1 Report

" A ∆E of 1.0 represents the smallest colour difference the human eye can see. Any ∆E less than 1.0 would not be noticeable, but a ∆E greater than 1.0 is usually noticeable. However, in certain circumstances a ∆E greater than 1.0 may not be noticeable, for example, two yellows of the same hue may have a ∆E greater than 1.0 but may not be noticeable. This is because of the saturation of the yellow colour [5]. But, most studies report ∆E<3 as clinically acceptable [14-16]."

Actually the ISO standard says that:

  • 0 < ΔE <1 – normal, invisible color variations
  • 1 < ΔE < 2 – small variations, recognizable only by an experienced observer
  • 2 < ΔE < 3.5 – medium variations, recognizable by inexperienced observer
  • 3.5 < ΔE < 5 – distinct color variations
  • ΔE > 5 – large color variations

Please provide the actual colors of materials. The Journal publishes papers with color figures and tables, so it would be very helpful and would visualize the described numerical results.

"Hemp fibre has more lignin and less cellulose in it, and dye cannot dye hemp fabric good. " - please provide more detailed explanation, beneficially regarding the chemistry of the process and the actual reason of such an effect.

Figure 4 is very hard to see. Please improve the quality, remove the numbers from the figure and put it in a separate table or separate spectra in figure. Also, provide the legend.

When describing changes in the mechanical performance please refer more to the actual changes in the chemical structure of material. Also schemes should be provided.

Author Response

Authors thank Reviewer for the careful analysis of our article and valuable remarks.

" A ∆E of 1.0 represents the smallest colour difference the human eye can see. Any ∆E less than 1.0 would not be noticeable, but a ∆E greater than 1.0 is usually noticeable. However, in certain circumstances a ∆E greater than 1.0 may not be noticeable, for example, two yellows of the same hue may have a ∆E greater than 1.0 but may not be noticeable. This is because of the saturation of the yellow colour [5]. But, most studies report ∆E<3 as clinically acceptable [14-16]."

Actually the ISO standard says that:

  • 0 < ΔE <1 – normal, invisible color variations
  • 1 < ΔE < 2 – small variations, recognizable only by an experienced observer
  • 2 < ΔE < 3.5 – medium variations, recognizable by inexperienced observer
  • 5 < ΔE < 5 – distinct color variations
  • ΔE > 5 – large color variations

Was added in hte text: „Usually according to ISO standard, 1 < ΔE < 2 represents small variations, recognizable only by an experienced observer.”

Please provide the actual colors of materials. The Journal publishes papers with color figures and tables, so it would be very helpful and would visualize the described numerical results. The Figure 4 with fabric samples was added.

"Hemp fibre has more lignin and less cellulose in it, and dye cannot dye hemp fabric good. " - please provide more detailed explanation, beneficially regarding the chemistry of the process and the actual reason of such an effect. Hemp fibre contains 5.68–7.96 % lignin and 64.7–75.38 % cellulose in it; flax fibre has 75.38–83.31 % cellulose and 2.21–4.78 % lignin [29]. Hemp yarns are rougher and have more remains of lignin than flax yarns [30]. Thus, these reasons can have the influence to fabrics dyebility, though chemical tests were not performed during this investigation. It is the problem of chemical technology of fibrous materials, and we have analysed the end-use properties of textile fabrics from these fibres.

Figure 4 is very hard to see. Please improve the quality, remove the numbers from the figure and put it in a separate table or separate spectra in figure. Also, provide the legend. Figure 4 was corrected: numbers were deleted, the legend was added and the quality improved as much as possible.

When describing changes in the mechanical performance please refer more to the actual changes in the chemical structure of material. Also schemes should be provided. Of course, dyeing process with reactive dyes is based on chemical reaction between dyestuff and fibre. Thus, it can have some influence to mechanical properties of woven fabric, but these changes are the object of fibrous materials chemical technology. We analyse just mechanical behaviour of textile.

Authors thank Reviewer for the careful analysis of our article and valuable remarks.

Reviewer 2 Report

Manuscript 1201929, titled “Mechanical and End-Use Properties of Cellulose and Cellulose/Protein Woven Fabrics” investigates the effect of dyeing on mechanical and end-of-use properties of woven fabrics made from cellulose and cellulose/protein woven.

The introduction, Materials and Methods as well as the Results section are very well written up until page 9, but after that the quality of the manuscript drops. Therefore I recommend consideration for publication after minor, but extensive, revisions, aimed at expanding discussion of the mechanical property results and revising some of data statistics. In addition, grammar and references need to be double-checked because there are many errors in the main manuscript, for example, citing the wrong reference number, etc. In addition, some of the language is not scientific. Expressions such as “changed very slightly” and “change a lot” are relative, and they need to be reworded.

The authors should address the following comments:

Main comments:

Comment 1: The title of the manuscript should more directly refer to the comparison of dyed and undyed fabrics, which was the main focus of this work. Readers cannot guess that by reading the current title. Please revise.

Comment 2: Abstract, lines 10-11, also Conclusions, line 486. The authors describe color changes as “very slightly”. This is relative and should be rephrased. For example “…for all fabrics DeltaE was smaller than 3, which is generally imperceptible to the human eye”.

Comment 3: Abstract, lines 16-17 also Conclusions, line 506. The authors write “Fabrics mass decreases during abrasion.” Does this need to be highlighted in the abstract? Generally, mass is lost during abrasion tests, therefore it is not a highlight of the presented research. I recommended removing this sentence.

Comment 4: The grammar and presentation of results of the “Results” section needs to be improved to make it easier to understand, particularly after line 350. Whereas the discussion of the optical properties is very thorough, the authors’ discussion of the crystallinity index and mechanical property results are very brief, almost superficial, and are lacking explanations why these results are what they are. If the results are unexpected please offer possible explanations based on literature or previous reports.

Comment 5: The authors repeatedly report the 95% confidence integral as the uncertainty value. Given the small sample size of the tested specimens (4 or 5 replicates per specimen cannot define a normal distribution) I recommend reporting the standard deviation and not the 95% confidence intervals. Same recommendation for the error bars associated with the plotted mechanical properties shown in Figures 5-10. 

Comment 6: Table 1: Please review standard EN ISO 105-J03 to verify appropriate number of significant digits reported. For example, should 1.77 +/-0.009 be reported as 1.77 +/-0.01 or 1.76X +/-0.0009? Also, the significant digits on the uncertainty value must match those of the reported (average) value. The uncertainty of a quantity cannot be more accurate that the quantity itself, as currently presented in Table 1. Please correct. For example, it is acceptable to report the following: 1.111+/-0.111 or 1.11+/-0.11 but not 1.11+/-0.111.

Comment 7: Table 2: The uncertainty for all measurements is the same (0.07). This is highly unlikely, except if it is due to rounding. Could the authors provide an explanation why the uncertainty values are all equal to 0.07? Also, see Comment 5, for use of standard deviation instead of standard error.

Comment 8: Table 3: Please check your error calculations. Error propagation for A-B is sqrt((error A)^2+ (error B)^2).

Comment 9: Table 4: Same comment regarding reported significant digits as Table 1.

Comment 10:  Line 387: The authors refer to Figure 3 clearly showing that all tested fabrics underwent a two step fading behavior. Figure 3 is in page 6 of the manuscript and does not show what the authors claim. Therefore the referenced figure is likely missing, please revise manuscript. In general, please refrain from using expressions such as “clearly shown” or “clearly shows”, and rephrase them. The readers do not need to be coaxed to clearly see results, if the results are there.

Comment 11: Lines 86-90: The paragraph discusses mechanical properties of fabrics, but references [1, 16] refer to mainly fabric color changes. Please revise with correct references.

Comment 12: Lines 60-68. Regarding the discussion of how the dyeing process affects fibre crystallinity/crystallite size, please add representative references.

Comment 13: Lines 350-355. Crystallinity index discussion. In the first sentence, the authors claim that the crystallinity indices of all tested fabrics are “very similar”. However, the listed fabric crystallinity indices are different. In order to back up the claim that the crystallinity indices are “similar” the authors should answer the following questions. How do variations in crystallinity index map to crystallinity obtained from X-ray diffraction measurements? In other words if the crystallinity index changes from 1.83 to 1.67, what is the actual reduction in crystallinity?

The last sentence discussing rigidity and flexibility seems more appropriate in the paragraphs discussing mechanical properties of fabrics, and should be moved in those paragraphs. What is the crystallinity index change for the rest of the fabrics reported in this work? Can the crystallinity index calculation from FTIR spectra be applied to those fabrics as well? If so, how does it correlate to mechanical property changes?

Comment 14: Lines 404-407. The authors offer a vague explanation as to why the silk/linen yarn becomes stronger when dyed. To strengthen their argument, they need to present more substantial evidence (other studies) that have observed similar effects in dyed silk/silk composites.

Comment 15:  Lines 481-484. The authors need to expand the discussion of Figure 11 results. Why is the mass loss of the silk/linen fabric the smallest for example? Do the linear fits to mass reduction provide information on the physical processes occurring during abrasion? Why is linear loss significant?

Comment 16: Conclusions, lines 506-507, “Linear equations describe dependencies exactly” Why is this important and what does it mean? This comment is similar to comment 15 above.

Comment 17: Conclusions section. Currently the Conclusions section has too many incoherent paragraphs. It should be revised to follow a more logical summary of the experimental results. Please summarize observations and briefly discuss what are the implications for of these observations. For example, the authors write “Fabrics area density increases from 18 to 36% after finishing”, line 502. Which fabric corresponds to what increase? (it was mentioned in the main text). Is this increase good or bad for the fabric? Also, the authors write “results change a lot”, lines 512. The language describing the change is subjective. Please revise accordingly.

Other comments:

Grammatical errors/formatting:

Line 10: replace “there was no” with “there were no”

Line 13: replace “have more crystalline” with “are more crystalline” or “have higher crystalline content”.

Line 57: replace “et al” with “et al.”

Line 80: Rathod and Kolhatkar is not found in the reference list and is not reference 15. Please correct.

Line 91: Please correct the Gabrijelcic reference number. Should be [19] not [2].

Line 93: Please correct the Gunaydin reference number. Should be [20] not [9].

Line 97: Please correct the Azeem reference number. Should be [21] not [17].

For all figure captions, please add “.” at the end of the caption.

For all table captions, please add “.” At the end of the caption.

Equations 1, 2, 3, 4 appear as a blurry images, please replace with word editor equations or higher resolution image.

Line 218: replace “FTIR spectras” with “FTIR spectra”.

Lines 219-220: replace “with the 4 cm-1” with “the 4 cm-1”.

Line 225: replace “was performed according” with “was calculated according”.

Line 238: replace “40 0C” with “40 °C”.

Line 239: Is the “ECE Reference Detergent” the “ECE reference detergent 77”?

Line 262: replace “cm2” with “cm2

Line 279: replace “warp direction..” with “warp direction.”

Lines 281-296: Change paragraph alignment to justified.

Line 315: replace “good” with “well”.

Line 319, replace “in a literature” with “in literature”.

Line 463, “fabric surface laughed,” what does it mean?

Reference formatting: Keep capitalization consistent for all references.

Generally: Proof read and revise all typographical errors, for example, in line 409 “mechnaical”. Replace all instances of “cm-1” with “cm-1” (-1 is a superscript). Also double-check cross-referencing of references within the main manuscript and the reference list at the end of the manuscript. As pointed out before some references in the main manuscript are incorrect. Correct all instances of “figure vs Figure”, “table vs Table”, etc.

Author Response

Authors thank Reviewer for the valuable remarks.

The introduction, Materials and Methods as well as the Results section are very well written up until page 9, but after that the quality of the manuscript drops. Therefore I recommend consideration for publication after minor, but extensive, revisions, aimed at expanding discussion of the mechanical property results and revising some of data statistics. In addition, grammar and references need to be double-checked because there are many errors in the main manuscript, for example, citing the wrong reference number, etc. In addition, some of the language is not scientific. Expressions such as “changed very slightly” and “change a lot” are relative, and they need to be reworded.

The authors should address the following comments:

Main comments:

Comment 1: The title of the manuscript should more directly refer to the comparison of dyed and undyed fabrics, which was the main focus of this work. Readers cannot guess that by reading the current title. Please revise. The title was changed into “Comparison of Mechanical and End-Use Properties of Grey and Dyed Cellulose and Cellulose/Protein Woven Fabrics”.

Comment 2: Abstract, lines 10-11, also Conclusions, line 486. The authors describe color changes as “very slightly”. This is relative and should be rephrased. For example “…for all fabrics DeltaE was smaller than 3, which is generally imperceptible to the human eye”. The changes were made.

Comment 3: Abstract, lines 16-17 also Conclusions, line 506. The authors write “Fabrics mass decreases during abrasion.” Does this need to be highlighted in the abstract? Generally, mass is lost during abrasion tests, therefore it is not a highlight of the presented research. I recommended removing this sentence. The recommended text was removed.

Comment 4: The grammar and presentation of results of the “Results” section needs to be improved to make it easier to understand, particularly after line 350. Whereas the discussion of the optical properties is very thorough, the authors’ discussion of the crystallinity index and mechanical property results are very brief, almost superficial, and are lacking explanations why these results are what they are. If the results are unexpected please offer possible explanations based on literature or previous reports. The comparison with other investigation was performed after each paragraph after line 350.

Comment 5: The authors repeatedly report the 95% confidence integral as the uncertainty value. Given the small sample size of the tested specimens (4 or 5 replicates per specimen cannot define a normal distribution) I recommend reporting the standard deviation and not the 95% confidence intervals. Same recommendation for the error bars associated with the plotted mechanical properties shown in Figures 5-10. The reproducibility and repeatability of the color measurement results among specimens of one sample is very high and the standard deviation of these measurements for all samples is about 0.08 (if rounded). Such standard deviation may be seen for almost all measurements of samples. As the measurement area is 9 mm and results obtained were highly replicated, so there is no need to carry out more measurements, because we will still receive similar results with similar standard deviation values. Regarding 95% confidence integral it is generally known, that for calculations of test results of textiles only 95% confidence integral may be used.

Comment 6: Table 1: Please review standard EN ISO 105-J03 to verify appropriate number of significant digits reported. For example, should 1.77 +/-0.009 be reported as 1.77 +/-0.01 or 1.76X +/-0.0009? Also, the significant digits on the uncertainty value must match those of the reported (average) value. The uncertainty of a quantity cannot be more accurate that the quantity itself, as currently presented in Table 1. Please correct. For example, it is acceptable to report the following: 1.111+/-0.111 or 1.11+/-0.11 but not 1.11+/-0.111. The results were corrected according to remark.

Comment 7: Table 2: The uncertainty for all measurements is the same (0.07). This is highly unlikely, except if it is due to rounding. Could the authors provide an explanation why the uncertainty values are all equal to 0.07? Also, see Comment 5, for use of standard deviation instead of standard error. As it was written next to the comment 5, the repeatability and reproducibility of colour measurement test results are very high. Yes, some results were rounded. If standard deviation values will be written next to the data of colour measurement results, 0.07 will be replaced to 0.08, but this will not provide more information.

Comment 8: Table 3: Please check your error calculations. Error propagation for A-B is sqrt((error A)^2+ (error B)^2). The confidence Interval formula was used:

X  ±  Z

s

√(n)

Where:

  • X is the mean
  • Z is the Z-value from the table (we took 1,960 at 95 % confidence)
  • s is the standard deviation
  • n is the number of observations

Comment 9: Table 4: Same comment regarding reported significant digits as Table 1. Corrections were made.

Comment 10:  Line 387: The authors refer to Figure 3 clearly showing that all tested fabrics underwent a two step fading behavior. Figure 3 is in page 6 of the manuscript and does not show what the authors claim. Therefore the referenced figure is likely missing, please revise manuscript. In general, please refrain from using expressions such as “clearly shown” or “clearly shows”, and rephrase them. The readers do not need to be coaxed to clearly see results, if the results are there. Figure 5 was added, numbers of other figures were corrected.

Comment 11: Lines 86-90: The paragraph discusses mechanical properties of fabrics, but references [1, 16] refer to mainly fabric color changes. Please revise with correct references. The references were corrected.

Comment 12: Lines 60-68. Regarding the discussion of how the dyeing process affects fibre crystallinity/crystallite size, please add representative references. The references were added.

Comment 13: Lines 350-355. Crystallinity index discussion. In the first sentence, the authors claim that the crystallinity indices of all tested fabrics are “very similar”. However, the listed fabric crystallinity indices are different. In order to back up the claim that the crystallinity indices are “similar” the authors should answer the following questions. How do variations in crystallinity index map to crystallinity obtained from X-ray diffraction measurements? In other words if the crystallinity index changes from 1.83 to 1.67, what is the actual reduction in crystallinity? The degree of crystallinity, the apparent lateral crystallite size, the proportion of crystallite interior chains and cellulose fraction increase with increasing of the wood density.  The hemp fibre is the roughest, but its crystallinity index is the lowest. Flax/silk fibre is the most flexible from these three fibres, but its crystallinity index is the medium. But still we can state, that flax fibers are more crystalline than hemp and that silk fibers are more amorphous and decreases crystallinity index of flax in flax/silk blended fabric.

The last sentence discussing rigidity and flexibility seems more appropriate in the paragraphs discussing mechanical properties of fabrics, and should be moved in those paragraphs. What is the crystallinity index change for the rest of the fabrics reported in this work? Can the crystallinity index calculation from FTIR spectra be applied to those fabrics as well? If so, how does it correlate to mechanical property changes? – the crystallinity index can be calculated to every fabric. It is clearly seen that; crystallinity index can be coherent to elongation at break of yarns or fabrics (see figures 7 and 9). Higher crystallinity index corresponds to lower elongation at break value. Higher crystallinity index shows that there are less amorphous areas in the fibres.

Comment 14: Lines 404-407. The authors offer a vague explanation as to why the silk/linen yarn becomes stronger when dyed. To strengthen their argument, they need to present more substantial evidence (other studies) that have observed similar effects in dyed silk/silk composites. This can be influenced by the fact that yarn with the part of elastic fibre stretched more after dyeing, its twist increased and yarn became stronger because of that. The tensile strength and elongation percentage of silk dyed yarns are found to be higher as compared to that of grey yarn [32].

Comment 15:  Lines 481-484. The authors need to expand the discussion of Figure 11 results. Why is the mass loss of the silk/linen fabric the smallest for example? Do the linear fits to mass reduction provide information on the physical processes occurring during abrasion? Why is linear loss significant? Figure 13 was changed into dependencies of mass loss on the number of abrasion cycles, which describes the abrasion process better.

Comment 16: Conclusions, lines 506-507, “Linear equations describe dependencies exactly” Why is this important and what does it mean? This comment is similar to comment 15 above. Linear equations described dependencies exactly, i.e. when the number of abrasion cycles increases, the mass loss also increases.

Comment 17: Conclusions section. Currently the Conclusions section has too many incoherent paragraphs. It should be revised to follow a more logical summary of the experimental results. Please summarize observations and briefly discuss what are the implications for of these observations. For example, the authors write “Fabrics area density increases from 18 to 36% after finishing”, line 502. Which fabric corresponds to what increase? (it was mentioned in the main text). Is this increase good or bad for the fabric? Also, the authors write “results change a lot”, lines 512. The language describing the change is subjective. Please revise accordingly. Conclusions were corrected.

The article was corrected by MDPI English Editor.

Authors thank Reviewers for the valuable remarks.

Round 2

Reviewer 1 Report

From my side everything is in order after corrections